# Understanding Antibiotic Resistance as a Perceived Threat towards Dairy Cattle through Beliefs and Practices: A Survey-Based Study of Dairy Farmers

**DOI:** 10.3390/antibiotics11080997

**Published:** 2022-07-25

**Authors:** Eleni Casseri, Ece Bulut, Sebastian Llanos Soto, Michelle Wemette, Alison Stout, Amelia Greiner Safi, Robert Lynch, Paolo Moroni, Renata Ivanek

**Affiliations:** 1Department of Population Medicine and Diagnostic Sciences, College of Veterinary Medicine, Cornell University, Ithaca, NY 14853, USA; eb643@cornell.edu (E.B.); sgl67@cornell.edu (S.L.S.); mmw85@cornell.edu (M.W.); aek68@cornell.edu (A.S.); pm389@cornell.edu (P.M.); ri25@cornell.edu (R.I.); 2Department of Public and Ecosystem Health, College of Veterinary Medicine, Cornell University, Ithaca, NY 14853, USA; alg52@cornell.edu; 3Department of Communication, College of Agriculture and Life Sciences, Cornell University, Ithaca, NY 14853, USA; 4College of Agriculture and Life Sciences, Cornell University, Ithaca, NY 14853, USA; rlynch@cornell.edu; 5Dipartimento di Medicina Veterinaria e Scienze Animali, Università degli Studi di Milano, Via dell’Università 6, 26900 Lodi, Italy

**Keywords:** antibiotic resistance, judicious antibiotic use, dairy farmer, questionnaire, culture-based mastitis treatment, attitudes

## Abstract

Antibiotic use is an important component in dairy herd management both to treat bacterial diseases and to maximize animal welfare. However, there is concern among scientists that antibiotic misuse and/or overuse by farmers might promote the emergence of resistant pathogens. We conducted a cross-sectional web-based questionnaire study with dairy farmers/managers in New York, USA to evaluate their (i) level of concern about antibiotic resistance and (ii) interest in adopting new judicious antibiotic use practices regarding mastitis treatment. A total of 118 responses were subjected to statistical analysis. The findings revealed that nearly half (45%) of study participants were undecided or disagreed that antibiotic resistance due to antibiotic use in dairy farming may negatively impact the health of dairy cattle. In contrast, the majority (78%) of participants self-reported that they do not treat with antibiotics at the first sign of mastitis, and the majority (66%) have either fully or partially implemented culture-based mastitis treatment on their farm. The self-reported adoption of culture-based mastitis treatment practices was statistically significantly associated with higher numbers of injectable and intramammary doses of antibiotics used on the participants’ farms. These findings will aid future research investigations on how to promote sustainable antibiotic use practices in dairy cattle.

## 1. Introduction

Antibiotics are used to treat and prevent bacterial infections in dairy cattle, including mastitis, one of the most common bacterial infections in dairy cows. It is estimated that 16% of all lactating dairy cows in the United States (US) receive antibiotic therapy for mastitis infections yearly. Additionally, 80% of dairy herds in the US have adopted blanket dry cow therapy for the prevention and cure of intramammary infections (i.e., administering doses of antibiotics via intramammary infusions following each lactation in every cow in all udder quarters) [1]. The most commonly used intramammary antibiotics are penicillins, cephalosporins, and other beta-lactams [2]. The resistance of bacteria to antibiotic agents is considered a public health threat globally, and antibiotic use in animal agriculture, including dairy operations, contributes to the burden of resistance [3]. However, the impact of antibiotic use in animal agriculture on the emergence and transmission of antimicrobial-resistant bacteria has not been fully understood due to the complexity of genetic dynamics involving resistance [4]. Even less is known about the threat of antibiotic resistance within the dairy industry, including the threat to cattle health. There is evidence of a widespread and emerging resistance to antibiotics by common mastitis pathogens (including *Staphylococcus* spp., *Klebsiella* spp. and pathogenic *Escherichia coli*) in the US and the world [5,6,7]. Given that there are multiple studies indicating the developing resistance of mastitis pathogens to the commonly used and available antibiotics as well as the spreading of resistant bacteria in the dairy industry, it is imperative that we are aware of the habits and beliefs of the dairy farmers and managers who make the decision to treat with antibiotics. This information is essential for understanding the drivers of antibiotic use in practice and for identifying potential motivators to change practices that are necessary for responsible antibiotic use and reduced antibiotic resistance in dairy farms.

Traditionally, recommendations for the management of mastitis in dairy farms have been about improvement of herd management practices such as the use of clean and dry beds for cattle housing, pre- and post-milking teat disinfection, sanitizing milking machines or feeding management (supplementation of vitamins, probiotics and probiotics) [8,9]. In the early 2010s, protocols for culturing mastitis pathogens became available and encouraged culture-based treatment in the US [8,10]. Today, antibiotic therapy is a well-established practice for the management of mastitis in dairy farms [8]. However, with only two classes of antibiotics approved for intramammary treatment in dairy cows in the US (β-lactams and one type of lincosamide), and no new intramammary antibiotics approved since 2006, it is important to ensure that the antibiotics available for farmers to treat mastitis infections continue to be effective, which makes the responsible use of antibiotics a core principle [11,12]. A 2020 interview study of New York dairy farmers and managers revealed that overall, producers feel that they use antibiotics in a way that qualifies as “judicious” but they have various levels of concern about antibiotic resistance affecting them at the farm level [13]. In our study, the objective was to further evaluate the attitudes of New York dairy farmers and managers towards antibiotic resistance and their day-to-day antibiotic use habits, including against mastitis infections. The purpose of doing this is to explore the reasons for antibiotic use in dairy farms and behaviors that protect both humans and animals from antibiotic-resistant infections.

## 2. Results and Discussion

A total of 1390 email addresses were sent the survey recruitment letter, representing 819 farms. Of these emails, 883 (63.5%) came from the database of Cornell PRO-DAIRY and 635 (45.7%) email addresses came from the database of the Cornell Quality Milk Production Services (QMPS) 128 (9.2%) email addresses were listed in both databases. A total of 131 dairy farmers/managers participated in the survey, but 13 participants who did not respond to questions beyond farm or demographic questions were dropped, leaving 118 participants for analysis. The response rate was 9.4% for the sample size at the individual farmers/manager level and 14.7% for the sample size at the farm level, while the completion rate was 79.4% (see Appendix A and Appendix A for details of the calculations). There were slightly more males (57%) that responded than females (Table 1). About half (51%) of the respondents were between the ages of 35–54 years old, and about one quarter (27%) were over the age of 55, and close to one quarter (21%) were 18–34 years old (Table 1). The vast majority (91%) of respondents plan to continue farming for 5 or more years (Table 1). Where possible, demographic variables (age, years farming, and gender identification) were compared to the 2017 US Department of Agriculture (USDA) Agriculture Census to confirm whether our survey participants represent the greater dairy farmer population. Age (*p =* 0.3) of our survey participants was not significantly different from those reported in the US census of dairy producers. However, gender composition (*p =* 0.001) and experience level (i.e., years farming, *p =* 0.0002) of the participants in our survey were significantly different from those reported in the US census. More survey participants described themselves as female compared to the statistics in the US census (43% of participants were female (Table 1) versus 30% identified in the US census), and survey respondents consisted of more inexperienced farmers and managers than those reported the US census (37% of participants have been farming for 10 years or less (Table 1) versus 21% in US census). In the previous month, participants’ farms had a median of 350.0 (interquartile range (IQR) 132.5–1313.5) lactating cows, had dried off a median of 26.0 (IQR: 10.0–85.0) cows, had a median of 1.7 (IQR: 0.9–3.3) cases of clinical mastitis/100 lactating cows, and had provided a median of 0.5 (IQR: 0.0–3.2) injectable and 1.9 (IQR: 0.0–5.3) intramammary doses of antibiotic per 100 lactating cows in their herd (not including dry cow treatments, Table 2).

In our survey, more than half of the survey respondents (55% of 102, Table 1) indicated that they believe that antibiotic resistance due to antibiotic use in dairy farming may negatively impact the health of dairy cattle (this question, Q17, will be referenced as “RESISTANCE BELIEFS” going forward). Just under half (45%) were either undecided or disagreed with the statement. This is noteworthy, as it indicates a lack of concern for, and potentially lack of awareness of, the problem of antibiotic resistance from almost half of the dairy farmers/managers represented in this study. These results are similar to a 2007 interview study of South Carolina dairy farmers, which revealed that overall, farmers did not seem concerned and even lacked awareness of how antibiotic resistance occurs, and even that viruses cannot be treated with antibiotics [14]. However, our results support the 2020 interview study of New York dairy farmers, where conventional dairy farmers interviewed did not rank antibiotic resistance as a high concern for human health but were more concerned about overuse causing the antibiotics to lose efficacy on their farm (i.e., they were concerned for the health of dairy cattle) [13]. Similarly, a literature review revealed that many dairy farmers in South Carolina, Washington and the UK were found to be aware that increased antimicrobial use can lead to antimicrobial resistance. The same literature review study also revealed that many Malaysian, Kenyan and Peruvian dairy farmers were aware that resistant bacteria could be more difficult to treat and posed a risk to their cattle [15]. Interestingly, the observed lack of concern or awareness in 45% of participants in our study about antimicrobial resistance affecting cattle was not explained by any of the evaluated explanatory variables (Appendix A). We hypothesize that this may be because (i) antibiotic resistance is not a tangible problem that they see affecting their day-to-day practices, and (ii) their antibiotic treatments usually work, especially when making culture-based treatment decisions, so they may feel that antibiotics not working due to resistance pathogens is not something that would affect them directly. This is in agreement with another interview study of New York dairy farmers, where some farmers expressed that when they do use antibiotics, they work well, so they do not feel antibiotic resistance is a threat to their operation [16]. Concluding from our results, it appears that among New York dairy farmers, there is room for education on the importance of prudent antibiotic use as it relates to the health of their cattle and potentially broader public health implications, as varying levels of antibiotic resistance within mastitis pathogens have been reported in dairy cattle both internationally in Canada and India and within the US [17,18,19]. Future educational initiatives may be necessary to familiarize US farmers on the rising threat of antibiotic resistance of common mastitis pathogens already occurring, both internationally and within the US.

Our study revealed that 66% of 108 respondents have either fully or partially implemented culture-based mastitis treatments on their farms (Table 1; this question, Q8, will be referenced as “CULTURE-BASED TREATMENTS going forward). Additionally, 16% self-reported that they would be willing to start carrying out culture-based mastitis treatments in the near future. This indicates that respondents use judicious antibiotic practices for mastitis treatment, as consistent culture-based mastitis treatments on farms can reduce (i) antibiotic loads on the farm by up to 50%, (ii) reduce the net cost of treatments while not reducing efficacy, and (iii) reduce time to cure by ensuring that the correct antibiotic is being used [10]. Culture-based treatments are also considered among “best practices” when developing a mastitis treatment protocol, along with good record-keeping and treatment protocol compliance among farmworkers, to ensure the best outcomes for mastitis cases [20]. The widespread practice of culture-based mastitis treatments among our study participants was surprising, and interestingly, it was not associated with the respondents’ concern about antibiotic resistance affecting the health of their cattle (Q17, “RESISTANCE BELIEFS”, Appendix A), suggesting that the concern about antibiotic resistance is not a motivator for their judicious antibiotic use practice. In addition to this variable, we evaluated other potential predictors for the binary version of the variable CULTURE-BASED TREATMENTS, which compared Uninterested participants (who are not interested in this practice) against Interested participants (who expressed any level of interest or performing this practice). Interestingly, in these analyses, the only statistically significant predictors of the interest in CULTURE-BASED TREATMENTS were the higher number of injectable doses of antibiotic per 100 lactating cows in the herd per month (Interested: median 1.0, IQR 0.0–3.7 vs. Uninterested: median 0.0, IQR 0.0–0.0; Mann–Whitney U test *p*-value 0.02) and the higher number of intramammary antibiotic doses (excluding dry cow treatments) per 100 lactating cows in the herd per month (Interested: median 2.0, IQR 0.0–5.7 vs. Uninterested: median 0.0, IQR 0.0–2.1; Mann–Whitney U test *p*-value 0.03) (Table 3). Additionally, the number of cases of clinical mastitis per 100 lactating cows in the herd (*p*-value 0.06) and the number of cows dried off (*p*-value 0.05) in the past month were borderline statistically significantly associated with the interest in CULTURE-BASED TREATMENTS. To better understand these results, we repeated this analysis with a categorical version of the CULTURE-BASED TREATMENTS variable that had three levels (Q8_3Levels): the same “Uninterested” level but the “Interested” level split into two groups of participants: those who are interested in culture-based treatments and are already using them (“Doing”) and those who are interested but not yet able or ready to use them (labeled “Interested”). This analysis revealed statistically significant differences between participants in the “Doing” and “Uninterested” levels for the number of cases of clinical mastitis per 100 lactating cows in the herd (Doing: median 2.1, IQR 0.9–3.8 vs. Uninterested: median 0.9, IQR 0.5–1.2; Dunn’s test Holm adjusted *p*-value 0.05) and the number of injectable doses of antibiotic per 100 lactating cows in the herd per month (Doing: median 1.2, IQR 0.0–4.1 vs. Uninterested: median 0.0, IQR 0.0–0.0; Dunn’s test Holm adjusted *p*-value 0.03). No statistical differences were observed between participants in the “Doing” and “Interested” or “Interested“ and “Uninterested” levels of the Q8_3Levels variable. Overall, these results are surprising, considering that the culture-based treatment has been touted as an approach to reduce the unnecessary use of antibiotics by more than 50% when used to treat mild and moderate clinical mastitis cases [10] However, because of the cross-sectional nature of our study, and the possibility of reversed causation, we cannot conclude anything about the presence and direction of the causal relationship between the participants’ interest in the culture-based treatments and their higher use of injectable and intramammary antibiotics. This should be the subject of future research.

Our results about the frequent implementation of the culture-based treatment mirror a 2016 study of Dutch dairy farmers [21] and their willingness to adopt selective dry cow treatments (SDCT), another proven way to reduce the amount of antibiotics used on dairy farms and reduce the risk of antibiotic-resistant pathogens from emerging. In that study, 75% of participating dairy farmers progressively took up SDCT on their farm during the study duration [21]. The willingness of farmers in our study to adopt culture-based mastitis treatments and the willingness of Dutch farmers to adopt SDCT indicates that widespread use of these judicious antibiotic use practices could be implemented. Moreover, we also revealed that the majority (78% of 91) of respondents indicated that they do not treat with antibiotics at the first sign of mastitis infection (Table 1; Q18 will be referenced as “FIRST SIGN MASTITIS” going forward). Thus, according to their self-reported survey responses, more than three-quarters of dairy farmers and managers represented in this survey study are using antibiotics appropriately for managing mastitis by waiting to treat mastitis and adopting management practices such as culture-based mastitis treatments. An important consideration for understanding farmers’ willingness or ability to adopt judicious antimicrobial use practices is understanding how to incentivize them. To that end, we found that the primary external incentives respondents would need in order to reduce antibiotic use on their farms (Q16) were financial incentives provided with their milk check (30% of respondents), grants to upgrade facilities to reduce infection risk (20%) and subsidized veterinary consulting/Quality Milk Production Services (16.7%) (Table 1). As farmers valued various incentives as equally important, the adoption of new practices or change in practices might require promising one or multiple benefits to dairy farmers and managers.

We knew dairy farmers and managers make daily decisions on whether they should administer an antibiotic treatment or not, but information was lacking about the specific motivating factors that play a factor in their decision-making process. For the respondents who do not treat with antibiotics at the first sign of mastitis infection, (answered “strongly disagree” or “disagree” to Q18 (FIRST SIGN MASTITIS)) we asked a follow up question (Q19 “FACTORS TO WAIT”) to better understand the reasons for waiting to give antibiotics and their motivations to use antibiotics (Figure 1). The respondents thought that “Culture Result” and “Severity of Health Problem” were extremely important when deciding to wait to treat mastitis infections with antibiotics. On the other hand, farmers considered factors relating to cost, labor, or withholding violations less important. These results slightly contradict a 2019 study of a geographically similar population (New York dairy farmers), in which surveyed farmers indicated that two out of the top three motivating attributes for using antibiotics prudently were “increased profitability” and “decreased risk of residues”, respectively [22]. However, our results do support the results of a 2021 focus group study of Canadian dairy farmers, who frequently cited severity of clinical signs in their animals as a factor in whether they should treat with antibiotics or wait, and were more likely to treat empirically than wait if the animal was febrile, for example [23].

Since Q19 (FACTORS TO WAIT) was only posed to farmers that answered a certain way on the question prior, one more question was explored that was presented to every survey-taker to better understand their motivations and antibiotic use habits. Question Q21 (FACTORS TO TREAT) revealed that respondents believe that “Animal welfare”, “Preventing disease spread” and “Trust that treating sooner with have a higher chance of a cure” are the most important reasons to treat at the first sign of mastitis infection (Figure 2). The only factor showing a more balanced distribution across response levels is “Convenience of quickly solving the problem with antibiotic treatment”. These specific results do support the 2019 study of the New York dairy farmers, where the top ranked motivating factor attributed for using antibiotics prudently was “health of the herd” [22].

The two heatmaps in Figure 1 and Figure 2 (for Q19 and Q21, respectively) suggest that dairy farmers’ and managers’ decisions to use or not use antibiotics are primarily based on keeping their cattle healthy, and other factors, such as cost or labor limitations, or limiting their antibiotic use are not as important. This supports the 2021 Canadian dairy farmer study, which found that farmers felt that cattle welfare is their responsibility, and they are not willing to jeopardize that by reducing antibiotic use [23]. Given that dairy cattle usually represent the entire livelihood of dairy farmers/managers, it is unsurprising that they find keeping their cattle healthy and free of disease very important and feel that utilizing antibiotics to help in this regard is appropriate. In summary, Q19 (FACTORS TO WAIT) and Q21 (FACTORS TO TREAT) suggest that the surveyed farmers make daily management decisions based on scientific evidence and maintaining high levels of animal welfare, and that labor, convenience and cost are not as critical as animal health.

Our study revealed that 66% of participants look to their herd veterinarians to learn about alternative treatments to antibiotics, and among those that utilize the farm veterinarian for information on antibiotic alternatives, 81% say the veterinarian is “very helpful” or “somewhat helpful” (Table 1). Currently, antibiotic alternatives available to dairy farmers include natural antibacterial peptides, biological response modifier products, pre- and pro-biotics, as well as ensuring adequate nutrition to maintain a healthy gut microbiome [24]. The role of herd veterinarians in advising dairy farmers about antibiotic alternatives adds to their already reported key role in helping dairy farmers (e.g., in the United Kingdom [25]) to implement judicious practices, reduce widespread antibiotic use on their farms, and increase farmer awareness of antibiotic resistance. Thus, utilizing veterinarians that regularly work with dairy farmers is a way to address the lack of concern and awareness among dairy farmers towards antibiotic resistance, as they find herd veterinarians to be helpful when advising them on antibiotic alternatives.

In our survey, 69.3% (63 out of 91) of the respondents stated that they would be interested in knowing how antibiotic use on their farm compares to antibiotic use on other similar dairy farms in New York (they either “strongly agree”, “agree”, or “somewhat agree” to the statement in Q15, Table 1). One implication suggested by this result is that future studies should target the development of a benchmark that would allow quantitatively measuring how much antibiotics are actually used on dairy farms for different treatment indications, modes of application, and active ingredients. This would support the ongoing research on antibiotic resistance, since multiple studies have focused on the quantification of antibiotic use in animal agriculture and recording of valid and comparable data on the antibiotic use at the farm and the national levels [26,27] including similar efforts that are already under way in Germany, where two methods of quantitatively measuring antibiotic usage on food animal production systems to establish benchmarks are being evaluated [28]. Our study indicated that the interest in benchmarking antimicrobial use is not limited to government agencies and scientists, but it also includes dairy farmers and managers. This is important because quantifying the actual antibiotic use is a fundamental requirement for evaluating the selection pressure for resistance and guiding decisions on antibiotic stewardship.

There are multiple study limitations that need to be acknowledged in light of drawing conclusions. First, the low response rate to the survey led to a very small sample size. Second, there could also be selection bias due to the source of participant emails; QMPS and PRO-DAIRY are both voluntary programs for dairy farmers to get involved in to improve their herd parameters and production, so the study participants may have higher rates of prudent antibiotic use and good habits compared to the larger population of dairy farmers in the target population of New York dairy farmers/managers. Third, with using email addresses versus physical addresses, Amish farms and farms run by older individuals, for whom email may be a novelty versus a necessity, were not included in the source population. Fourth, our survey sample was representative of the US dairy producers in terms of age distribution of the participants; however, there were more female and inexperienced participants (farming for less than 5 years) in our survey compared to in the US census, which could have introduced selection bias into the study findings. There could also be information bias, due to relying on information self-reported by participants, which may not be entirely accurate. While we planned to control for the confounding bias through multivariable regression, no multivariable model was built presenting a lack of evidence of confounding (whether true or due to the low power), however, there could have been undetected confounding due to unmeasured variables. Future investigations should prospectively collect quantitative data about the use of specific antibiotic classes on dairy farms to make more accurate conclusions on farmer and manager antibiotic use habits.

## 3. Conclusions

Nearly half of the study participants self-reported lack of concern about antibiotic resistance as a threat to dairy cattle health on their farms. Nevertheless, many of the surveyed farmers are already doing culture-based mastitis testing and wait to treat with antibiotics until culture results are available, along with farmers reporting that they use the severity of the mastitis infection to determine if antibiotics should be administered. Interestingly, the self-reported adoption of culture-based mastitis treatment practices was statistically significantly associated with higher numbers of injectable and intramammary doses of antibiotics used on the participants’ farms, but the direction of causal relationship, if any, could not be determined in this cross-sectional study. Overall, study findings provide preliminary data for future educational and research initiatives to aid efforts to promote sustainable antibiotic use practices in dairy cattle based on improved understanding of farmer and manager beliefs and practices.

## 4. Methods

### 4.1. Survey Instrument

A web-based cross-sectional survey was conducted between August and November 2019. The survey was developed using Qualtrics (https://www.qualtrics.com/, accessed on 24 May 2022), a free survey software, and consisted of 30 questions, with 21 multiple choice questions and 9 open ended/free text questions. The questions collected information about participant demographics, farm characteristics, disease prevention strategies, alternatives to antibiotics, farmer opinions on optimizing antibiotic use and antibiotic resistance, mastitis treatment decisions, and more in-depth questions about farm characteristics. The survey was developed by two co-authors (A.S., R.I.) and then sent to others (A.G.S., R.L., P.M.) for feedback and revision. A pilot survey was further sent to an ambulatory clinician at Cornell Ambulatory and Production Medicine service, and their suggestions were implemented to improve understandability and response rate. The final survey was therefore sent out once the questionnaire was finalized. The complete survey instrument (“Farm management survey”) can be found in the Appendix A.

### 4.2. Survey Recruitment

The source population selected for our study consisted of the customers of Cornell Quality Milk Production Service (QMPS) and members of the Cornell PRO-DAIRY whose farms are located in the Northeastern US (majority New York State, with some in bordering Northeastern states due to the wide range of QMPS clients). To recruit dairy farmers/managers from within this source population, we distributed the survey through email lists of QMPS customers and Cornell PRO-DAIRY members. Cornell QMPS is a service that tests milk samples for farmers to assess their milk quality, antibiotic residues and mastitis. Cornell PRO-DAIRY is a research program dedicated to facilitating the advancement of New York’s dairy industry through working with farmers across the state on industry-applied research. The recruitment email inviting farmers to participate in the survey was sent to email addresses on the three lists, i.e., to dairy farmers/managers in the QMPS database only, those in the PRO-DAIRY database only, and to farmers/managers who were listed in both of these databases. To be able to track individuals from the three lists, we created three identical surveys and for each list we used a separate survey link. The initial recruitment email was sent on 20 August 2019 and was followed by three reminder emails on 17 September, 7 October, and 30 October 2019. The first recruitment email bounced back for some addresses, which was attributed to the inability of Outlook to manage a large number of email addresses. Thus, it is possible that many addresses were not effectively contacted through the initial recruitment email. However, this was successfully resolved by sending the reminder emails to smaller batches of email addresses. The recruitment email briefly described the survey, contained a link and QR code to take the survey on Qualtrics, and provided information on the incentive (USD 5 electronic gift card for Carhartt). If the farmers chose to take the survey, they were prompted with a consent form to agree to before they could fill out the survey. Except for the exclusion due to incomplete answers (explained below), all dairy farmers/managers who voluntarily filled out the survey were included in the study.

### 4.3. Data Analysis

*Data cleaning and descriptive statistics.* The data were cleaned and analyzed throughout the study using R Studio (https://www.rstudio.com/, accessed on 24 May 2022). Data cleaning consisted of removing responses from 13 participants who did not respond to questions beyond farm or demographic questions. We calculated response rate and completion rate for our survey based on the guidelines provided by the American Association for Public Opinion Research (AAPOR) [29] (Appendix A and explanations in Appendix A). We used two types of sample sizes to estimate the response rate: (1) sample size at individual level and (2) sample size at farm level. For the individual level, sample size was equal to the total number of email addresses/individuals the survey was sent to (1390). The farm level sample size is the total number of farms represented by our survey sample (890; i.e., each farm was represented by 1.6 emails/individuals on average, or one farm may have been represented by one or more individuals). The farm level response rate assumes that a single representative of a farm responded to the survey. For the analysis, first each survey question’s responses were summarized with figures and summary statistics. Bar plots were used for categorical variables and scatterplots were used for numerical variables. For the two questions (Q19 and Q21) where survey participants ranked factors on the level of importance, results from those questions were summarized through generating heatmaps. Demographic characteristics of the survey participants were compared to the US census of dairy producers estimated by the USDA in 2017 with a one-sample z test [30]. The demographic characteristics included in these comparisons were age, gender and years farming.

#### 4.3.1. Univariable Analysis

Two categorical outcomes of interest were chosen for this study; Q17 (Level of agreement with the statement, *“Antibiotic resistance due to antibiotic use in dairy farming may negatively impact the health of dairy cattle.”* (referred to as “RESISTANCE BELIEFS”) and Q8 (Level of interest in adopting culture-based treatment for mastitis cases (referred to as “CULTURE-BASED TREATMENTS”). When grouping response levels in Q17, all responses for “Strongly Disagree”, “Disagree”, and “Undecided” were categorized as “Disagree”, and “Strongly agree” and “Agree” were categorized as “Agree”. When grouping Q8, all responses for “I am not interested in this” was labeled as “Uninterested”, while “I am interested but unable”, “I would be ready to do this in the near future”, “I am already doing aspects of this” and “I am already doing this fully” were categorized as “Interested”. Associations were visually analyzed by generating bar plots and balloon plots for each potential predictor against the outcomes of interest. Tests of associations between each of the two binary outcomes of interest and all other survey questions (predictors), one at a time, as potential predictors, were carried out using Chi-square Tests or Fisher’s Exact Tests (as applicable) if the variable was categorical, or the Mann–Whitney U Test if the variable was continuous. In addition to the binary version of Q8, we created a categorical version of this variable with three levels (Q8_3Levels) to be able to assess any differences among (i) participants uninterested in doing culture-based treatments, (ii) participants interested in culture-based treatments but not yet doing this, and (iii) participants who are already doing at least some aspects of this practice. Therefore, as in the binary version of this question, the response “I am not interested in this” was labeled as “Uninterested”, however, “I am interested but unable” and “I would be ready to do this in the near future” were labeled as “Interested”, while the responses “I am already doing aspects of this” and “I am already doing this fully” were categorized as “Doing”. An association of this Q8_3Levels variable with continuous explanatory variables was assessed using the Kruskal–Wallis test followed by pairwise multiple comparisons with Dunn’s Test and Holm adjustment of *p*-values for multiple comparisons. Associations with the membership in the QMPS, PRO-DAIRY or both source populations were assessed using a Chi-Square Test (Q17) and Fisher’s Exact Test (Q8) to determine whether participants’ membership in these groups may be affecting the study findings, but no associations were found, so all responses from both groups were combined for analysis. In all analyses, a statistical significance threshold of 5% was used.

#### 4.3.2. Multivariable Analysis

Among variables evaluated in the univariable analyses only two were found to be statistically significantly associated with one of the outcomes of interest (Q8) (at the 5% significance threshold) but no multivariable model could be built. Hierarchical clustering (using “cluster” package in R) was used to determine whether participants group in a certain way based on their responses to survey questions; however, no conclusions were drawn from this analysis because of the small sample size.

## Figures and Tables

**Figure 1 antibiotics-11-00997-f001:**
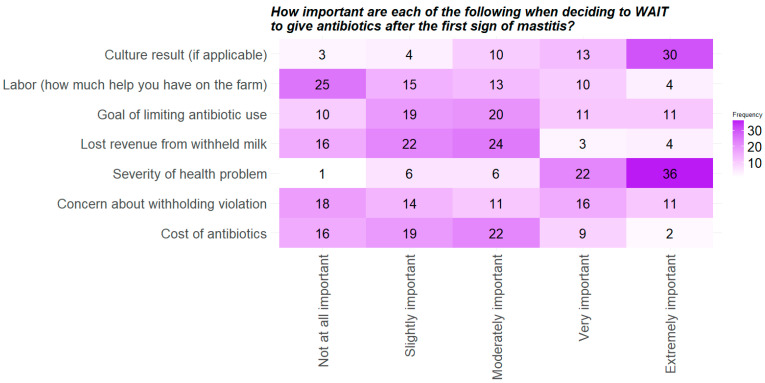
Heatmap of responses Q19, “How important are each of the following when deciding to WAIT to give antibiotics after the first sign of mastitis?” This question was only displayed to 72 out of 91 respondents who answered either “Strongly disagree” or “Disagree” to Q18 (FIRST SIGN MASTITIS). Farmers ranked each factor individually along the scale from “Not at all important” to “Extremely important.”.

**Figure 2 antibiotics-11-00997-f002:**
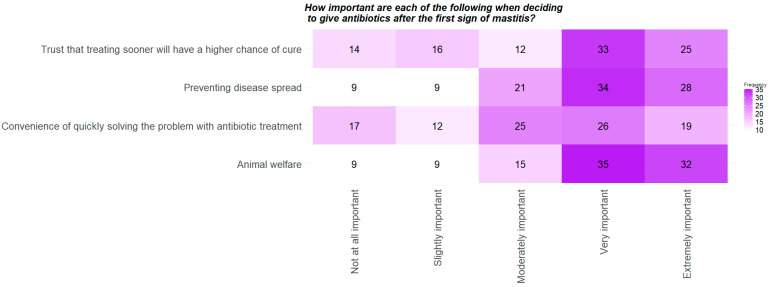
Heatmap of Q21, “How important are each of the following when deciding to give antibiotics at the first sign of mastitis?” Farmers ranked each factor individually along the scale from “Not at all important” to “Extremely important.” All farmers were displayed this question.

**Table 1 antibiotics-11-00997-t001:** Descriptive statistics for the responses to the categorical survey questions in the order they appeared in the survey instrument ^a^.

No.	Question (Total Responses for the Question) Answer Choices	Frequency	Percent
	**Tell us about your farm**		
**Q2**	**What most accurately describes your farm operation? (117)**		
	Conventional	105	89.7
	Organic	12	10.3
**Q3**	**What is your position/role on the dairy farm? (117)**		
	Herd manager	24	20.5
	Owner/co-owner	85	72.6
	Other	8	6.8
**Q4**	**How long have you owned or managed your farm? (109)**		
	Less than 1 year	3	2.8
	Up to 5 years	18	16.5
	Between 5 and 10 years	19	17.4
	Between 10 and 20 years	27	24.8
	**Disease prevention practices**		
**Q8**	**How interested are you in adopting culture-based (pathogen based) testing for treating mastitis? Culture based testing involves taking a milk sample from a cow showing clinical mastitis in order to help identify the cause and best course of treatment.** **(108)**		
	I am not interested in this	14	13
	I am interested but unable ^b^	5	4.6
	I would be ready to do this in the near future	17	15.7
	I am already doing aspects of this	35	32.4
	I am already doing this fully	37	34.3
**Q10**	**How interested are you in constructing a new barn or making similar significant changes to your farm facility to improve herd management? (108)**		
	I am not interested in this	10	9.3
	I am interested but unable ^c^	31	28.7
	I would be ready to do this in the near future	42	38.9
	I am already doing/have done this fully	25	23.1
	**Alternatives to antibiotics**		
**Q12**	**Which of the following product attributes is the most important if you needed to choose an antibiotic alternative to treat your lactating cows? (102)**		
	Ease of administration	8	7.5
	Scientifically proven	50	47.2
	Cost of the product	19	17.9
	Milk withhold time	13	12.3
	Allowed under organic certification	9	8.5
	Other ^d^	7	6.6
**Q13**	**Who or what sources do you consult the most often to learn about alternatives to antibiotics? (106)**		
	Herd veterinarian	70	66
	Cooperative Extension/Quality Milk Production Services	2	1.9
	Dairy publications, such as Hoard’s Dairyman	10	9.4
	Nutritionist	0	0
	Milk inspector	0	0
	Other dairy farmers	11	10.4
	None, I am not interested in antibiotic alternatives	5	4.7
	Other ^e^	8	7.5
**Q14**	**How helpful is your veterinarian in advising you about alternatives to antibiotics? (107)**		
	Not helpful	13	12.1
	Somewhat helpful	41	38.3
	Very helpful	46	42.9
	Unsure	7	6.5
	**Optimizing antibiotic use**		
**Q15**	**State your level of agreement: “*I would be interested in knowing how antibiotic use on my farm compares to use on other similar dairy farms in NY*.” (91)**		
	Strongly agree	12	13.2
	Agree	36	39.6
	Somewhat agree	15	16.5
	Neither agree nor disagree	26	28.6
	Somewhat disagree	1	1.1
	Disagree	1	1.1
	Strongly disagree	0	
**Q16**	**What primary external incentive would you need in order to reduce antibiotic use on your farm? (90)**		
	Financial incentives provided with your milk check	27	30
	Grants to upgrade facilities to reduce infection risk	18	20
	Subsidized veterinary consulting/Quality Milk Production Services	15	16.7
	Tax incentives	0	
	Other ^f^	9	10
	None of the above	21	23.3
**Q17**	**Please state your level of agreement with the following statement: “*Antibiotic resistance due to antibiotic use in dairy farming may negatively impact the health of dairy cattle.”* (102)**		
	Strongly agree	13	12.7
	Agree	43	42.2
	Undecided	25	24.5
	Disagree	14	13.7
	Strongly disagree	7	6.9
	**Mastitis treatment decisions**		
**Q18**	**State your level of agreement: “*I treat with antibiotics at the first sign of mastitis infection*.” (91)**		
	Strongly agree	3	3.3
	Agree	16	17.6
	Disagree	44	48.4
	Strongly disagree	28	30.8
	**Demographic questions**		
**Q5**	**What is your age? (95)**		
	18–24	2	2.1
	25–34	18	18.9
	35–44	32	33.7
	45–54	17	17.9
	55+	26	27.4
**Q6**	**What is your gender? (94)**		
	Female	40	42.6
	Male	53	56.3
	Prefer not to say	1	1.1
**Q7**	**What is the highest level of education you have completed? (95)**		
	Post graduate training or professional school	6	6.3
	College graduate (Bachelor’s degree)	56	58.9
	Trade school, associate’s degree, or some college	22	23.1
	High school graduate or GED	11	11.6
**Q37**	**How long do you plan to continue farming? (94)**		
	Less than 5 years	8	8.5
	More than 5 Years	86	91.5

^a^ The question (Q) number notation matches that in the Qualtrics dataset. Summary statistic for Question Q22 “If a cow could maintain its organic status, would you consider using antibiotics on your farm?” is not provided because of a typo in the question choices, which made the responses invalid. ^b^ The 5 participants who selected “I am interested but unable” in Q8 were offered Question Q9 “If unable to adopt culture-based testing for treating mastitis, what is the primary barrier?”; their responses were: Financial reasons (1), Labor issues (1), No access to a lab for testing (2) and Other (1). ^c^ The 31 participants who selected “I am interested but unable” in Q10 were offered Question Q11 “If unable to construct a new barn or make similar significant changes to your farm facility, what is the primary barrier?”; their responses were: Financial reasons (29) and Other (2; their free choice responses were “milk coop limitations” and “milk market quota-financial”). ^d^ The 7 participants who selected “Other” in Q12 in their free choice response stated: “Efficacy”, “efficacy”, “Effectiveness of the antibiotic!!!”, “We don’t treat for mastitis”, “Most effect for the given situation”, “is it naturopathic”. ^e^ The 8 participants who selected “Other” in Q13 in their free choice response stated: “QMPS”, “qmps”, “online”, “staff vet at organic valley”, “The Complete Herbal Handbook for Farm and Stable”, “other organic farmers”, “Besides prevention with vaccination not aware of proven alternatives”. ^f^ The 9 participants who selected “Other” in Q16 in their free choice response stated: “Improved benchmark/health performance”, “Any of the above”, “already reducing antibiotic use”, “research and education”, “We only use dry treat for antibiotics”, “Proof that cows would be healthier”, “Something truly equally effective”, “Proof that there are viable and quick acting alternatives”.

**Table 2 antibiotics-11-00997-t002:** Descriptive statistics for all responses to the numerical free choice survey questions about the participants’ farms and calculated frequencies of (i) clinical mastitis and (ii) injectable and (iii) intramammary antibiotic treatment doses per 100 lactating cows ^a,b^.

Question No. ^c^	Question (Total Responses for the Question)	Min	1st Quartile	Median	3rd Quartile	Max
Q23	What was the average number of lactating cows in your herd for the last month? (91)	5.0	132.5	350.0	1313.5	3700.0
Q31	How many cows did you dry off last month? (91)	0.0	10.0	26.0	85.0	300.0
Q24	Over the past month, how many cases of clinical mastitis did you experience on your farm? (89)	0.0	2.0	6.0	15.0	173.0
Q25	How many injectable antibiotic treatments have been provided to lactating cows on your farm in the last month? An antibiotic treatment is defined as a single dose, administered to a single animal. Please provide the total number of doses given across all lactating cows over the last month. (85)	0.0	0.0	2.0	16.0	250.0
Q26	How many intramammary antibiotic treatments have been provided to lactating cows on your farm in the last month (not including dry cow treatment)? An antibiotic treatment is defined as a single dose, administered to a single animal. Please provide the total number of doses given across all lactating cows over the past month. (86)	0.0	0.0	5.5	30.0	150.0
(Q24/Q23) × 100 ^d^	Cases of clinical mastitis per 100 lactating cows (88)	0.0	0.9	1.7	3.3	12.6
(Q25/Q23) × 100 ^d^	Doses of injectable antibiotics per 100 lactating cows (84)	0.0	0.0	0.5	3.2	26.7
(Q26/Q23) × 100 ^d^	Doses of intramammary antibiotics per 100 lactating cows (85)	0.0	0.0	1.9	5.3	28.6

^a^ Question (Q32) “What was your average somatic cell count last month?” is not included due to discrepancies in how the numerical values were entered by the participants, and therefore statistical analysis was not possible. ^b^ Participant with ID 105 was removed from this analysis because of an invalid response to Q24 (response entered was 0.02) and their response to Q23 was interpreted as a typo (response entered was 1). ^c^ The question (Q) number notations match those in the Qualtrics dataset. ^d^ Frequencies were calculated by normalizing the participants’ responses to Questions Q24, Q25 and Q26 by the number of lactating cows on their dairy farm (Q23).

**Table 3 antibiotics-11-00997-t003:** Descriptive statistics and the results of univariable analysis of the association between the binary version of the outcome of interest Q8 (Uninterested vs. Interested in adopting culture-based testing for treating mastitis, referenced as “CULTURE-BASED TREATMENTS”) and variables representing the numerical free choice survey questions as well as variables describing the calculated frequency of (i) clinical mastitis and (ii) injectable and (iii) intramammary antibiotic treatment doses per 100 lactating cows ^a,b^.

	Q8: Uninterested					Q8: Interested					*p*-Value ^d^
Question No. ^c^	Min	1st Quartile	Median	3rd Quartile	Max	Min	1st Quartile	Median	3rd Quartile	Max	
Q23 ^e^	5.0	18.8	82.5	1125.0	3200.0	6.0	150.0	400.0	1330.0	3700.0	0.09
Q31 ^f^	0.0	0.0	0.0	50.0	300.0	0.0	10.0	30.0	97.8	300.0	0.05
Q24 ^g^	0.0	1.0	2.0	15.0	20.0	0.0	3.0	6.0	17.0	173.0	0.20
Q25 ^h^	0.0	0.0	0.0	0.0	16.0	0.0	0.0	3.0	19.3	250.0	0.01
Q26 ^i^	0.0	0.0	0.0	3.5	45.0	0.0	0.0	9.0	40.0	150.0	0.02
(Q24/Q23) × 100 ^j^	0.0	0.5	0.9	1.2	3.3	0.0	0.9	1.9	3.4	12.6	0.06
(Q25/Q23) × 100 ^j^	0.0	0.0	0.0	0.0	3.3	0.0	0.0	1.0	3.7	26.7	0.02
(Q26/Q23) × 100 ^j^	0.0	0.0	0.0	2.1	3.5	0.0	0.0	2.0	5.7	28.6	0.03

^a^ Question (Q32) “What was your average somatic cell count last month?” is not included due to discrepancies in how the numerical values were entered by the participants, and therefore statistical analysis was not possible. ^b^ Participant with ID 105 was removed from this analysis because of an invalid response to Q24 (response entered was 0.02) and their response to Q23 was interpreted as a typo (response entered was 1). ^c^ The question (Q) number notations match those in the Qualtrics dataset. ^d^ Mann–Whitney U test. ^e^ Q23: “What was the average number of lactating cows in your herd for the last month?” ^f^ Q31: “How many cows did you dry off last month?” ^g^ Q24: “Over the past month, how many cases of clinical mastitis did you experience on your farm?” ^h^ Q25: “How many injectable antibiotic treatments have been provided to lactating cows on your farm in the last month? An antibiotic treatment is defined as a single dose, administered to a single animal. Please provide the total number of doses given across all lactating cows over the last month.” ^i^ Q26: “How many intramammary antibiotic treatments have been pro-vided to lactating cows on your farm in the last month (not including dry cow treatment)? An antibiotic treatment is defined as a single dose, administered to a single animal. Please provide the total number of doses given across all lactating cows over the past month.” ^j^ Frequencies were calculated by dividing the participants’ responses to Questions Q24, Q25 and Q26 by the number of lactating cows on their dairy farm (Q23).

## Data Availability

Cleaned dataset that was used in the analyses is available in a publicly accessible repository (https://github.com/IvanekLab/AMR-Farmer-Survey, accessed on 24 May 2022).

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
