# Peer review of "Understanding Antibiotic Resistance as a Perceived Threat towards Dairy Cattle through Beliefs and Practices: A Survey-Based Study of Dairy Farmers"

_antibiotics, 2022, doi:10.3390/antibiotics11080997_

Round 1

Reviewer 1 Report

The multidrug resistance is a major global issue, currently, we see a huge increase in drug resistance among the microorganisms and importantly. The persistent and tolerant microorganisms are the major driving factors leading to the development of multi-drug resistant.

So the prevention of this type of infections is crucial, starting from the animal’s health.

I have some suggestions and worries. 

1. the manuscript dos not present many details about the antibiotic resistance. Please give us details obout thet.

2. Can you details the classes of antibiotics that were used for the treatment. It will be useful. 

3. please add some more criteria of selection and exclusion for your study.

4. add 3 short conclusions.

Thank You! 

Author Response

Response to reviewers

The authors thank the reviewers for their thoughtful comments. All comments have been addressed. Under each reviewer comment we describe and quote (in bold) the changes that have been made in the manuscript in response to the comment. The indicated line numbers correspond to the new/changed text in the revised manuscript viewed with tracked changes in the “All Markup” mode. We believe that the requested changes have resulted in an improved manuscript and hope you will find the revised manuscript acceptable for publication.

Reviewer 1:

Comments and Suggestions for Authors 

The multidrug resistance is a major global issue, currently, we see a huge increase in drug resistance among the microorganisms and importantly. The persistent and tolerant microorganisms are the major driving factors leading to the development of multi-drug resistant.

So the prevention of this type of infections is crucial, starting from the animal’s health.

I have some suggestions and worries. 

  1. the manuscript dos not present many details about the antibiotic resistance. Please give us details obout thet.

Response: Thank you for raising this point. We added a section in the first introduction paragraph to address your comment and included details about antibiotic resistance. The section is as follows: “Resistance of bacteria to antibiotic agents is considered a public health threat globally, and antibiotic use in animal agriculture, including dairy operations, contributes to the burden of resistance [3]. However, the impact of antibiotic use in animal agriculture on the emergence and transmission of antimicrobial-resistant bacteria has not been fully understood due to the complexity of genetic dynamics involving resistance [4]. Even less is known about the threat of antibiotic resistance within the dairy industry, including to cattle health. There is evidence of a widespread and emerging resistance to antibiotics by common mastitis pathogens (including Staphylococcus spp., Klebsiella spp. and pathogenic Escherichia coli) in the US and the world [5], [6], [7]. (lines 52-64)

  1. Can you details the classes of antibiotics that were used for the treatment. It will be useful. 

Response: We agree that information about the antibiotic classes used in treatments by participating farmers and managers would be useful. However, we are concerned whether a web-based survey, like the one conducted here, would result in information on antibiotic classes of sufficient quality and accuracy, considering the possibility of recall bias and lack of opportunity to interact with the participants and clarify questions. Instead, we believe that a field study that prospectively collects quantitative data about antibiotic use would be a more appropriate choice for a study design to obtain that information. We clarified that in the study limitations: “Future investigations should prospectively collect quantitative data about the use of specific antibiotic classes on dairy farms to make more accurate conclusions on farmer and manager antibiotic use habits.” (lines 381-384)

  1. please add some more criteria of selection and exclusion for your study.

Response: Thank you for this feedback. The source population selected for our study consisted of the customers of Cornell Quality Milk Production Service and members of the Cornell PRO-DAIRY whose farms are located in the Northeastern U.S. The study population (participants) were dairy farmers/managers from this source population who volunteered to participate. We clarified this in the Methods – Survey Recruitment section “The source population selected for our study consisted of the customers of Cornell Quality Milk Production Service (QMPS) and members of the Cornell PRO-DAIRY whose farms are located in the Northeastern U.S. (majority New York State, with some in bordering Northeastern states due to the wide range of QMPS clients). To recruit dairy farmers/managers from within this source population, we distributed the survey through email lists of QMPS customers and Cornell PRO-DAIRY members.” (lines 430-436) and “Except for the exclusion due to incomplete answers (explained below), all dairy farmers/ managers who voluntarily filled out the survey were included in the study.” (lines 454-456)

  1. add 3 short conclusions.

Response: We revised our conclusion section by summarizing 3 most important findings: “Nearly half of the study participants self-reported lack of concern about antibiotic resistance as a threat to dairy cattle health on their farms. Nevertheless, many of the surveyed farmers are already doing culture-based mastitis testing and wait to treat with antibiotics until culture results are available, along with farmers reporting they use the severity of the mastitis infection to determine if antibiotics should be administered. Interestingly, the self-reported adoption of culture-based mastitis treatment practices was statistically significantly associated with higher numbers of injectable and intramammary doses of antibiotics used on the participant’s farm, but the direction of causal relationship, if any, could not be determined in this cross-sectional study. Overall, study findings provide preliminary data for future educational and research initiatives to aid efforts to promote sustainable antibiotic use practices in dairy cattle based on improved understanding of farmer and manager beliefs and practices.” (lines 396-413)

Thank You! 

Response: Thank you for all your comments! We hope our revisions have resolved the raised issues. Should there be additional suggestions we would be happy to consider them.

Reviewer 2 Report

Introduction:

1. Would like to have the information on when the introduction of culture-based treatment before giving cows antibiotics in US was and types of interventions/sensitizations among farmers were conducted so far. So that recommendation for future educational initiatives can be more clarified in conclusion.

2. Revise introduction regarding the antibiotic resistance of mastitis causing pathogens because the citied paper is of 2012. Worldwide, Escherichia coli, Staphylococcus spp. and Klebsiella spp. resistance to common antibiotics is increasing

Results and Discussion:

1. Table 3 was taken time to interpret since it was required to get back to Table 2 to identify what is the questions of Q23, Q31, etc. It could be also helpful if frequency of these binary groups is indicated in the Table 3 instead of referring 434 to find out details.

2. Line 434, grouping Q8, among 5 answers, standing alone one answer and categorised other 4 answers into one group is questionable. Resulting in distribution of 13% vs. 87%. Within this 87%, there are 2 types; “giving antibiotics based on culture results” and “giving antibiotics without getting culture results”. If farmers giving antibiotics based on culture results, it is rationale. Would like to know the contribution of high doses from later groups, no results but use antibiotics.

3. Questionnaire is not asking much on antibiotic resistance (Q17 only) and all other questions are regarding with beliefs and practices of antibiotic use focusing on adopting culture-based treatment. Results shows association of adopting culture-based treatments and use of antibiotics but not supported by data to explain motivation of adoption is whether for the prevention for antibiotic resistance. Therefore, the title of understanding antibiotic resistance… could be too broad.

Author Response

Response to reviewers

The authors thank the reviewers for their thoughtful comments. All comments have been addressed. Under each reviewer comment we describe and quote (in bold) the changes that have been made in the manuscript in response to the comment. The indicated line numbers correspond to the new/changed text in the revised manuscript viewed with tracked changes in the “All Markup” mode. We believe that the requested changes have resulted in an improved manuscript and hope you will find the revised manuscript acceptable for publication.

Comments and Suggestions for Authors

Introduction:

  1. Would like to have the information on when the introduction of culture-based treatment before giving cows antibiotics in US was and types of interventions/sensitizations among farmers were conducted so far. So that recommendation for future educational initiatives can be more clarified in conclusion.

Response: Thank you for this valid comment. We added a section in the introduction to provide information about the culture-based treatments of mastitis and treatments to prevent mastitis in dairy cattle: “Traditionally, recommendations for the management of mastitis in dairy farms have been about improvement of herd management practices such as use of clean and dry beds for cattle housing, pre- and post-milking teat disinfection, sanitizing milking machines or feeding management (supplementation of vitamins, probiotics and probiotics) [8},[9].. In the early 2010s, protocols for culturing mastitis pathogens became available and encouraged culture-based treatment in the US [8],[10].”(Lines 68-73)

  1. Revise introduction regarding the antibiotic resistance of mastitis causing pathogens because the citied paper is of 2012. Worldwide, Escherichia coli, Staphylococcus spp. and Klebsiella spp. resistance to common antibiotics is increasing

Response: Thank you for this feedback. We revised the sentence according to your suggestion and cited recent articles (published in 2020 and 2022). The new sentence in the manuscript reads as follows: “There is evidence of a widespread and emerging resistance to antibiotics by common mastitis pathogens (including Staphylococcus spp., Klebsiella spp. and pathogenic Escherichia coli) in the US and the world [5],[6],[7]]” (Line 49-53).

Results and Discussion:

  1. Table 3 was taken time to interpret since it was required to get back to Table 2 to identify what is the questions of Q23, Q31, etc. It could be also helpful if frequency of these binary groups is indicated in the Table 3 instead of referring 434 to find out details.

Response: Thank you for this valid point. We agree that Table 3 should be self-standing and should not require Table 2 for interpretation. Accordingly, we included the wording of each question in the footnotes of Table 3 (lines 256-265). We avoided inserting a separate column for including the wording of each question (like in Table 2) to eliminate crowding in the table and provide large enough area for the frequencies of the groups. We opted to not indicate the frequency of the two outcome groups (Uninterested and Interested) for each explanatory variable in Table 3 because that would require adding two additional columns to the already busy table. However, we are open to other suggestions if this is still a concern. 

  1. Line 434, grouping Q8, among 5 answers, standing alone one answer and categorised other 4 answers into one group is questionable. Resulting in distribution of 13% vs. 87%. Within this 87%, there are 2 types; “giving antibiotics based on culture results” and “giving antibiotics without getting culture results”. If farmers giving antibiotics based on culture results, it is rationale. Would like to know the contribution of high doses from later groups, no results but use antibiotics.

Response: Thank you for this valuable comment; it helped us further understand our results. Accordingly, as per your suggestion, we regrouped the respondents of Q8 into 3 groups based on their self-reported interest in culture-based treatments and whether they are already doing this if interested: (i) Uninterested, (ii) Interested (but not yet doing this) and (iii) Doing (interested and doing this already). This analysis revealed that both “number of cases of clinical mastitis per 100 lactating cows in the herd” and “the number of injectable doses of antibiotic per 100 lactating cows in the herd per month” were statistically significantly higher for the Doing group compared to Uninterested group, which further explained a similar results for the binary version of Q8.

The detailed methods for the analysis is provided in Methods section in lines 495-506as follows: “In addition to the binary version of Q8, we created a categorical version of this variable with 3 levels (Q8_3Levels) to be able to assess any differences among (i) participants uninterested in doing culture-based treatments, (ii) participants interested in culture-based treatments but not yet doing this, and (iii) participants who are already doing at least some aspects of this practice. Therefore, as in the binary version of this question, the response “I am not interested in this” was labeled as “Uninterested”, however, “I am interested but unable” and “I would be ready to do this in the near future” were labeled as “Interested”, while the responses “I am already doing aspects of this” and “I am already doing this fully” were categorized as “Doing”. An association of this Q8_3Levels variable with continuous explanatory variables was assessed using the Kruskal Wallis test followed by pairwise multiple comparisons with Dunn’s Test and Holm adjustment of p-values for multiple comparisons.”

And our results regarding this analysis is provided in lines 222-235 as follows: “To better understand these results, we repeated this analysis with a categorical version of the CULTURE-BASED TREATMENTS variable that had three levels (Q8_3Levels): the same “Uninterested” level but the “Interested” level split into two groups of participants: those who are interested in culture-based treatments and are already using them (“Doing”) and those who are interested but not yet able or ready to use them (labeled “Interested”). This analysis revealed statistically significant differences between participants in the “Doing” and “Uninterested” levels for the number of cases of clinical mastitis per 100 lactating cows in the herd (Doing: median 2.1, IQR 0.9-3.8 vs Uninterested: median 0.9, IQR 0.5-1.2; Dunn’s test Holm adjusted p-value 0.05) and the number of injectable doses of antibiotic per 100 lactating cows in the herd per month (Doing: median 1.2, IQR 0.0-4.1 vs Uninterested: median 0.0, IQR 0.0-0.0; Dunn’s test Holm adjusted p-value 0.03). No statistical differences were observed between participants in the “Doing” and “Interested” or “Interested“ and “Uninterested” levels of Q8_3Levels variable.”

  1. Questionnaire is not asking much on antibiotic resistance (Q17 only) and all other questions are regarding with beliefs and practices of antibiotic use focusing on adopting culture-based treatment. Results shows association of adopting culture-based treatments and use of antibiotics but not supported by data to explain motivation of adoption is whether for the prevention for antibiotic resistance. Therefore, the title of understanding antibiotic resistance… could be too broad.

Response: Thank you for your comment. It is correct that most of the survey questions were about beliefs and practices about antibiotic use. However, the question about perceptions of antibiotic resistance (Q17) is one of our two main outcome variables of interest and, thus, we believe that mentioning antimicrobial resistance in the title is appropriate. Still, to more precisely reflect the scope of our study by conveying that our study is specifically about perceptions, we revised the title as follows (the newly added word is underlined):  “Understanding antibiotic resistance as a perceived threat to-wards dairy cattle through beliefs and practices: a survey-based study of dairy farmers. ” (lines 2-4). Thank you for all your comments.